# Method and Practice for Integrated Water Landscapes Management: River Contracts for Resilient Territories and Communities Facing Climate Change

Francesca Rossi 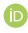

Department of Planning, Design, Technology of Architecture, Sapienza University of Rome, 00196 Rome, Italy; fra.rossi@uniroma1.it; Tel.: +39-339-7113348

**Abstract:** The negative impacts of climate change on natural and anthropic ecosystems have led to the increasingly urgent search for policies, strategies and tools able to counteract degradation and risk factors on vulnerable landscapes. Among these, the research activity refers to water landscapes as a specific field of study that represents a fundamental resource for human well-being. In consistency with the international policy framework on integrated water management, this contribution develops a case study analysis focused on River Contracts as innovative, voluntary and negotiated planning practices aimed at the reconstruction of territorial, social and ecological values, which broaden the boundaries of safeguarding by integrating protection actions with sustainable management and environmental regeneration and to restore the identity of places and local communities. The description and evaluation of an ongoing experience of River, Coast and Landscape Contracts, developed along the coast of the Lazio region, allows one to point out that the process method is successful in solving complex issues related to the management of the river basin while involving social actors in order to improve people's knowledge of the territory, increase social awareness of risk conditions, and allow local communities to propose and implement shared solutions. The results of this territorial and multi-level governance method are therefore measured on their capacity to produce territorial, social and environmental resilience.

**Keywords:** water landscapes management; river contracts; climate change; resilience

## 1. Introduction

### 1.1. Water Landscapes: A Shared Framework to Face Climate Change

The international debate on the management and conservation of the world's natural resources began in the early 1990s, and it highlighted the importance of water as a resource to be protected as a crucial environmental element of a territory [1–3]. In 2000, with the Water Framework Directive (WFD) 2000/60/EC, a focus for community action to maintain and improve the quality of water resources throughout Europe was defined. Water was recognized as a crucial resource for the sustainable development of local communities through the integration of human needs, the maintenance of aquatic ecosystems and the mitigation of the effects of floods and droughts.

In particular, the directive underlined the need to use integrated management systems for water and neighboring territories, whose governance and control policies must be flanked by other environmental and land management policies in order to pursue quality objectives by coordinating and integrating the disciplines involved in the knowledge process and enhancement of responsibilities, legislation and measures as well as through the involvement of institutions and citizens. The common objective to achieve, by 2015, the "good status" for the quality of waters in a hydrographic district was set together with the general protection of the aquatic ecology, the specific protection of unique and valuable habitats, and the protection of drinking water resources and of bathing water. All these

objectives had to be integrated for each river basin, while ecological protection should apply to all waters, according to the central requirement of the directive, which underlines that the environment needs to be protected to a high level in its entirety [4].

With the establishment of a common framework for the assessment and reduction of flood risks by the European Parliament in 2007, member states have been asked to reduce the adverse consequences for human health, the environment, cultural heritage and related economic activities by favoring a long-term planning approach in order to integrate all the cognitive data on hazards, vulnerabilities and hydraulic risks [5].

The activities of the European Environmental Agency also confirmed European river-fronts and coastlines to be affected by widespread and progressive degradation, in terms of habitat loss, eutrophication, contamination and erosion, recalling, with the reports, "*Blueprint to Safeguard Europe's Water Resources 2012*" and "*Climate change, impacts and vulnerability in Europe 2016*", the strategic objective of ensuring the availability of good quality water to meet the needs of citizens and to support the economy and the environment, through the integration of water policy objectives into other related policy areas such as agriculture, fisheries, renewable energy and transport [6–8].

The threat posed by climate change, the progressive erosion of natural resources, the polluting effects related to private motorization, energy waste, excessive consumption of agricultural and natural soils and their progressive waterproofing and reduction in vegetation cover, and the subsequent environmental degradation have highlighted, even more urgently, the need for a sustainable management of natural resources in terms of an equitable and ethical development to build inclusive communities and be able to adapt to the negative effects of emergency phenomena through new capacities of economic, cultural and social resilience [9,10].

At the same time, the restoration of ecosystems according to less aggressive methods in terms of consumption, pollution and fragmentation of the ecological-environmental connections, together with the protection of biodiversity by proactively responding to the fragility and vulnerability of environmental risks, represent a fundamental operation for ecological regeneration and historical-cultural and socio-economic valorization. The identity of places and local communities, recognized in water landscapes for their significant heritage and cultural value, can enable communities to contrast the increasingly widespread dynamics of abandonment and degradation. In fact, the vulnerability of these territories, where the coexistence of different living beings is defined in terms of "reciprocity", identifies the conditions and opportunities for a new territorial articulation in which ecology restores to the landscape its performative function, making it a new territorial infrastructure to support all new transformation and development activities [11].

The Water Framework Directive, the Sustainable Development Goals of the 2030 Agenda of the United Nations, the New Urban Agenda, the European Green Deal and the Biodiversity Strategy for 2030 are the main implementation references to conserve and promote a sustainable use of water resources for sustainable development and a comprehensive, ambitious and long-term plan to protect nature and reverse the degradation of ecosystems [12–14]. In this framework, the implementation of the River Contract as an innovative instrument, basically voluntary, of strategic and negotiated planning, pursues the protection, the correct management of water resources and the valorization of river territories, together with the safeguarding of hydraulic risk, and so contributes to local development [15,16].

*1.2. River Contracts: An Innovative, Voluntary Practice for Water Landscapes' Regeneration*

The Document of the 2nd World Water Forum held in The Hague in 2000 introduced River Contracts as instruments that allow to adopt criteria of public utility, economic performance, social value and environmental sustainability to intervene equally in the search for effective solutions [17]. A demonstration that the achievement of integrated water resource management needs an effort that is not only institutional, but first and foremost cultural, so that waters can be perceived and governed as "landscapes of life" [18].

A cultural approach is clearly reflected in the mentioned international policies on water resources, which also adopt the community principles of democratic participation in decision-making with the implementation of a territorial participatory process that fully captures the "regional and local dimension". Moreover, in accordance with the United Nations Conference on Sustainable Development (Rio+20, Rio de Janeiro, 2012), River Contracts are declined as instruments aimed at experimenting with a new governance system for sustainable development through an integrated approach between development and environmental protection policies, with the emphasis set on the urgency of characterizing actions on both an international and local scale, focusing on the fundamental role of participation and co-responsibility in the decision-making processes in development choices [19].

River Contracts have developed throughout Europe, starting in France in the early 1980s and then spreading within a few years to many other nations, such as Belgium, Luxembourg, the Netherlands, Spain, Italy and, in many cases, in the form of cross-border processes involving multiple territories. In France, contracts de rivière were promoted by the Interministerial Committee to support local initiatives designed to curb water quality degradation, control pollution and flooding, manage hydraulic structures and raise stakeholder awareness about supporting the management of water resources to increase the quality of life [20]. The first River Contract was signed in 1983, covering the La Thur basin. The results achieved by the French experiences are an important reference for the implementation and use of innovative management modalities as well as the experiences outside of Europe, in particular the River Contract in Canada, especially in Canada and in Africa

In Italy, where the entire framework described has been assimilated into the national legislation quite slowly [21], the first contract to be signed was in Lombardy regarding the Rivers Olona-Bozzente-Lura in 2004. The contents of the Water Framework Directive have been formally implemented with the Legislative Decree 152/2006—"Environmental Regulations". Afterwards, River Contracts were introduced with the Law no. 221/2015, "Dispositions on environmental matters to promote green economy measures and for the containment of the excessive use of natural resources", also known as the "Environmental Attachment" to the Stability Law, with the introduction of a new article into the Legislative Decree 152/2006, the 68 bis, entitled to them.

River Contracts were then mentioned in "Annex-3: Proposals for action" of the National Adaptation to Climate Change Strategy (SNAC) [22] to encourage participatory forms of water resource management as adaptation actions to climate change in the short term and in the long term at the level of river basins or individual water bodies, confirming the directives of the International Panel on Climate Change (IPCC) [23] and of the European Environmental Agency (EEA) on Italy's vulnerabilities in the context of the Mediterranean area, caused by extreme weather phenomena. Therefore, River Contracts, were identified as innovative tools for the potential and protection of landscapes and cultural heritage, able to create resilient communities and territories facing natural and anthropogenic risks.

To overcome the fragmentation between levels of Government in Italian Regions and to promote local participation for the management and enhancement of rivers, the National Board on River Contracts defined them as voluntary agreements of strategic and negotiated planning aimed at the protection, proper management, recovery and enhancement of river territories. They are also aimed at flood protection, the mitigation of hydrogeological risk, the implementation of the environmental role of agriculture, the development of ecological production areas, proper land use and the integrated protection of water resources, based on the involvement of local communities in making decisions on river basin management [24].

Finally, in 2018, the Decree of the Minister of the Environment and Protection of Land and Sea (MATTM) established a National Observatory of River Contracts at MATTM with functions of direction and coordination for the harmonization and application of River Contracts in Italy.

*1.3. River, Lake, Coastal and Lanscape Contracts: A Resilient and Multi-Level Governance Method*

In Italy, River, Lake and Coastal Contracts are configured, according to the National Charter of River Contracts, as continuous processes of negotiation aimed at the containment of eco-landscape degradation and the redevelopment of river basins/sub-basins and their territories and communities. The process of the contracts is determined in accordance with the different regulatory frameworks, the peculiarities of the basins, and in correlation to the needs of the territories and in response to the needs and instances expressed by the citizens, and is guided by three principles:

- Horizontal and vertical subsidiarity;
- Participatory local development;
- Sustainability.

The principle of subsidiarity is fostered by horizontal collaboration on a local scale between administrations, citizens and associations and vertical coordination between institutions (municipalities, mountain communities, parks, provinces, regions, basin/district authorities, the state and the European Union) in order to overcome the difficulties caused by the fragmentary nature of institutional and territorial competences. Subsidiarity is also a foundational element of the Integrated Water Resource Management (IWRM) paradigm, promoting coordinated development and management of water, land and related resources in order to maximize economic and social welfare in an equitable manner without compromising the sustainability of vital systems [25].

The principle of participation is intended as a driver for local development based on the awareness and sense of belonging of local communities to the basin as a matrix of its own cultural identity. According to an eco-systemic approach and as indicated in the Water Framework Directive, participation is the only interrelation modality capable of capturing the territorial identity and transferring its distinctive features into the strategic choices of local development in order to effectively achieve the objectives of valorization and protection (WFD, art. 14 2000/60/EC).

The principle of sustainability pursues the balance between ecology, equity and economy for the development of territories without threatening the operability of the natural, built and social systems on which the provision of environmental, social and economic services depends. The wide literature on the topic, starting from the Bruntland Report (World Commission on Environment and Development, 1987), finds an exhaustive reference for understanding the strong relationship between sustainability indicators, resilience and climate change in more recent IPPC reports [26].

In accordance with the above principles, the methodology of River Contracts is defined in the Action Program, which contributes at different scales (European, national, regional and local) to achieve, through the activation of a bottom-up participatory process, concrete and lasting results in respect of the territorial, social and administrative context with specific reference to the objectives of the programs and plans already in force in the territory. In the National Chart of River Contracts, the eight phases of the River Contracts methodology are defined as follows:

1. Phase 1: Document of Intent (Manifesto). The vision and general objectives, with reference to the Directive 2000/60/EC (Art. 4), point out the specific critical issues and the working methodology, shared among the actors taking part in the process.
2. Phase 2: Integrated Knowledge Analysis. The environmental, social, economic and cultural values of the territory.
3. Phase 3: Strategic Plan. The scenario, referred to as a medium-long-term time horizon, that integrates the objectives of local and territorial planning.
4. Phase 4: Action Program (AP). To define the time horizon (from three to five years) of the activities and update the contract with the approval of a new AP.
5. Phase 5: Participatory Processes. To enable the sharing of intentions, commitments and responsibilities among the stakeholders.

6. Phase 6: Formal Agreement. To define the decisions shared in the participatory process and the specific commitments of the contractors.
7. Phase 7: Periodic Monitoring System. To check the status of implementation of the various phases and actions, the quality of participation and the resulting deliberative processes.
8. Phase 8: Public Information. Accessibility to data and information on the River Contract.

This process methodology is also recalled in the document "Definitions and basic quality requirements of River Contracts", established in 2015 by the National Table of River Contracts, the Ministry of Environment and Land and Sea Protection (MATTM) and the Italian Institute for Environmental Protection and Research (ISPRA) [27].

Referring to this framework, this contribution, as part of the activities of an ongoing research project entitled "Regenerating the coastal territories of the Middle Tyrrhenian Sea. Landscape itineraries for resilient communities along the coast and in the minor islands of Lazio" funded by Sapienza, University of Rome, proposes a reflection on water landscapes as ecological devices that, starting from the potential of places, can contribute to reconstruct, through integrated and inter-scalar actions, processes of attachment and belonging to restore the existing heritage and make it a principle of identity for those who inhabit it.

The research activity described in the following paragraphs has focused on the Lazio Region River Contracts to investigate the contribution given to the development and implementation of this significant field of study.

## 2. Materials and Methods

### 2.1. River Contracts in the Lazio Region: An Ongoing Research Study

In accordance with the perspective and objectives underlined in the previous phase of contextualization, the research has deepened the ongoing processes in the Lazio Region and then developed a case study analysis, focusing on the description and evaluation of a specific cases of River, Coast and Landscape Contracts to highlight the existing relationship between the objectives of the River Contracts and the ecological, economic and social objectives at the base of sustainable development of the territorial contexts in which they are localized. Among the 20 projects going on in the Lazio Region, the selected case study concerns a very significant territorial context, localized in the Province of Rome: the *River, Coast and Landscape Contract Arrone*. The selection of the case study has been made according to its geographical location along the coastal system of the region stretching from the north to the proximity of Rome and is the expression of the many environmental characteristics of the territorial development of the Lazio coast. The contract also represents an exhaustive sample of the contract's procedure, with a very well-structured database of information available on the web. The contract is still in process, but it has already completed many of the foreseen steps, according to the National Charter of River Contracts.

The study also focuses on the description and evaluation of the results (Section 3) of the River, Coast and Landscape Contract Arrone to recognize, with an inductive method, the territorial, social and environmental resilience-oriented objectives and actions implemented in the process (*Phases 1–5*, Cfr. Par.1.3). In the Discussion and Conclusion section (Section 4), the potentials and limits related to the formalization, monitoring and dissemination of the contracts (*Phases 6–8*, Cfr. Par.1.3) are highlighted to introduce the open issues and further research questions in terms of implementation and diffusion of the procedures. The research methodology assumes the mentioned main guiding principles of River Contracts (horizontal and vertical subsidiarity, participatory local development and sustainability (cfr. Par.1.3)) as a reference for the correspondence of the process with the national framework, and searches to verify and evaluate the River, Coast and Landscape Contract Arrone capacity to produce resilience in terms of territorial, social and ecological processes of adaptation, learning and development [28]. A concept of resilience is intended not only as a response to disturbances but also as a flexible strategy towards eco-system renewal, according to which river basins can be considered as interesting laboratories for being drivers of territorial policies for ordinary communities and landscapes, enhancing the relationship between the

sustainable use of territorial resources and leading to new territorial strategies. Moreover, the "promotion of managements synergies" at different levels of regional and local planning, based on social participation, ensures a resilient planning approach to which River Contracts can contribute in order to produce and enhance resilience in river territories and to maintain their basic functions and structures at times of disturbance [29–31].

### 2.2. The Lazio Region River Contracts a Recent Field of Experimentation

The Lazio Region adhered to the National Charter of River Contracts (National Table of River Contracts 2010) in 2014, and in 2016, the Regional Law 17/2016 recognized the River Contract as "*a voluntary strategic and participatory planning tool, aimed at the integrated management of river basin and sub-basin policies, at the protection, enhancement and redevelopment of water resources and related environments, at the safeguard against hydraulic risk, at the sustainable management of the naturalness and landscape of rivers and hydrogeological risk, contributing to the local development of the interested territories*" (Art. 3 para. 95 Regional Law no. 17 of 31 December 2016).

In 2018, the Lazio Region established a purpose office called "Small Municipalities and River Contracts" to support the activities regarding the enhancement of river territories, and in 2019, the "Regional Forum of River, Lake, Mouth and Coastal Contracts" to foster an informative/consultative comparison between the region and the representatives of the River Contracts, through a "Technical Table of River, Lake, Mouth and Coastal". These actions were aimed at providing the regional coordination for different types of contracts within European, national and regional planning, identifying new forms of funding and priority actions for the protection of the territory through participatory processes involving local actors. In the same year, the Lazio Region started the drafting of an "Atlas of Objectives" with the aim of making a territorial, environmental and landscape reference framework available to the participatory processes and, in particular, to deepen the awareness of the territories interested by the River, Lake and Coastal Contracts in the region.

Intended to favor a greater knowledge of the already existing plans and programs potentially or directly affecting the choices of the contracts, the atlas is structured to be a representative/descriptive document of the existing and foreseeable territorial transformations, declined by territorial areas, contributing to the sharing of addresses and measures that allow River Contracts, started and being started at a regional level, to reach common objectives concerning water quality, soil protection, hydraulic safety, eco-systemic quality and agricultural enhancement, landscape and environmental requalification and valorization and economic development [32].

By 2020, 20 projects had been funded by the Lazio Region according to evaluation criteria based on the level of implementation of the process, the involvement of local actors, the environmental, landscape and historical-cultural features of the contract, the possibility of co-financing, the project quality in terms of clarity and completeness (i.e., planned actions, definition of objectives and degree of attainability).

In 2022, eight procedures have succeeded in signing the Formal Agreement: the River Contract Tevere from Castel Giubileo to the mouth, the River Contract Media Valle del Tevere, the River Contract Aniene, the Coastal Contract Agro Pontino, the River Contract Ufente, the Lake Contract Bracciano, the Coastal Contract Riviera di Ulisse and the River Contract Paglia (Table 1 and Figure 1).

### 2.3. The Contextualisation of the Case Study

In accordance with the research activities focused on in the Lazio region, the selection of the case study moves from some preliminary considerations about water landscapes as particular ecosystems in which the strong connections existing between the river basin and its coastal surroundings make even more evident the delicate environmental transition between rural, marine and fluvial ecosystems. A transition that configures a particularly complex condition in terms of biodiversity, ecosystem service productivity and natural and anthropic vulnerability.

**Table 1.** River Contract (RC), Lake Contract (LC), Coastal Contract (CC), River Coast and Landscape Contract (RCLC) and Coast and Mouth Contract (CMC) in the Lazio Region. Source: authors' elaboration (2022) from data of the national board on River Contracts available at http://www.a21fiumi.eu/ (accessed on 30 June 2022).

| River, Lake, Coastal Contract | Document of Intent (Phase 1) | Formal Agreement (Phase 6) |
|---|---|---|
| RCLC Arrone | signed in 2020 | ongoing |
| CC Agro Pontino | signed in 2019 | signed in 2022 |
| CMC Riviera di Ulisse | signed in 2019 | signed in 2022 |
| LC Bolsena | signed in 2017 | ongoing |
| LC Bracciano | signed in 2019 | signed in 2022 |
| LC Paola * | signed in 2018 | ongoing |
| RC Almone | signed in 2016 | ongoing |
| RC Amaseno | signed in 2018 | ongoing |
| RC Aniene | signed in 2018 | signed in 2022 |
| CC Cavata Linea Pio | signed in 2015 | ongoing |
| RC Cosa | signed in 2017 | ongoing |
| RC Farfa | signed in 2019 | ongoing |
| RC Fibreno | ongoing | ongoing |
| RC Garigliano Basso Liri | signed in 2016 | ongoing |
| RC Melfa | signed in 2017 | ongoing |
| RC Paglia | signed in 2014 | signed in 2022 |
| RC Rio Capodacqua | signed in 2019 | ongoing |
| RC Sacco | signed in 2015 | ongoing |
| RC Tevere (C. Giubileo to Mouth) | signed in 2017 | signed in 2022 |
| RC Medio Tevere (Farfa) | signed in 2014 | signed in 2022 |
| RC Ufente | signed in 2018 | signed in 2022 |

* The Lake Contract Paola is included in the territory of the Coastal Contract Agro Pontino.

The impact of environmental risks on these particular landscapes and, in particular, of the hydrological effects caused by flooding combined with resource erosion, pollution, energy waste and land consumption make them strongly altered and alterable from a structural, morphological and ecological point of view.

In this framework, the Lazio coast represents a unique environmental, historical, cultural and urban continuum, structured on a multiplicity of paths and testimonies that have a centuries-old history of land, maritime, commercial and tourist exchanges, where the diversity and specificity of local contexts are integrated in a complex and unitary system in which natural and semi-natural areas intersect the urban settlements. A territory whose landscapes, heterogeneous in their geographical characteristics, correspond to a more unitary cultural identity to which belongs a common historical heritage. A palimpsest that substantiates the permanence and importance of the settled communities as well as the scenario of a progressive condition of vulnerability [33]. These delicate territories, highly infrastructured and crossed by transport and trade lines, suffer from a multiplicity of territorial and socio-economic contradictions, including the strong pressure of seasonal tourism, the depletion processes of available natural resources, and the depopulation processes during the winter months. Moreover, the progressive abandonment of agricultural and maritime traditional activities, has progressively led to a loss of local identity and of cultural and landscape values [34].

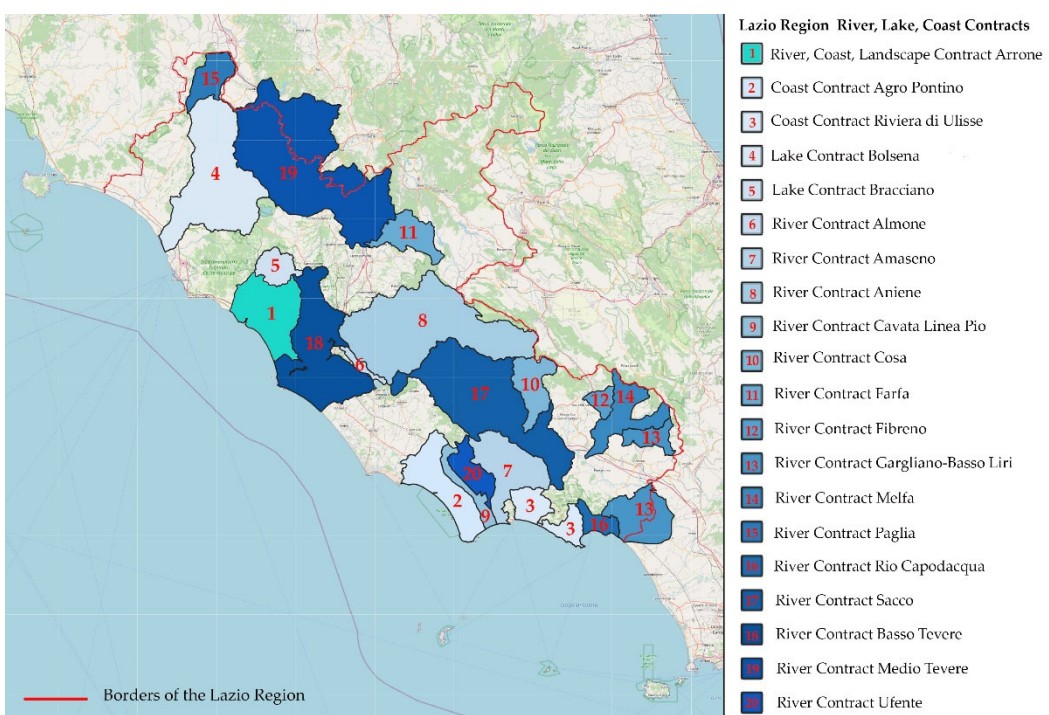

**Figure 1.** The localization of the Lazio Region River, Lake, Coastal Contracts, highlighting the case study of the River, Coast and Landscape Contract Arrone. Source: author's elaboration (2022) from data of the River Contract Atlas available at https://progetti.regione.lazio.it/contrattidifiume/atlante-dei-cdf/ (accessed on 30 June 2022).

The selected case study, the River, Coast and Landscape Contract Arrone, represents a valid example aimed at the production of territorial, social and environmental resilience to sustain local communities as they face climate change through the implementation of mitigation and adaptation actions. It concerns a particular coastal landscape in which the relationships between natural and anthropic environments are strongly weakened by uncontrolled soil consumption and increasing tourist pressure.

At the same time, the studied territory includes a multitude of naturally protected areas for the many ecological values and cultural resources that need to be safeguarded and enhanced.

The River, Coastal and Landscape Contract Arrone covers the territory north of the mouth of the Tiber, and is closely connected with the rural territory of the Agro Romano, and constitutes a landscape in which the natural and anthropic structure, stratified over the centuries, still appears strongly recognizable in the regular pattern of cultivated fields, in the course of the drainage canals and riparian vegetation and in the territorial structure defined by the system of agricultural villages.

The municipalities of Fiumicino and Cerveteri, to which this area mainly belongs, are two of the largest agricultural districts in Italy, characterized by the stratification of different production models that make it an emblematic example of the development of Italian agriculture. Places that were originally inhospitable due to their marshy and unhealthy soils and sparsely inhabited, where the first attempts at land reclamation date back to the Etruscans. The site reached its greatest development under the power of Rome as part of the Roman agricultural quadrilateral that stretched from the Argentario to Capua, between the Tyrrhenian Sea and the Apennines [35].

With the reclamation of Maccarese, Campo Salino, Ostia and Isola Sacra, this portion of the Agro was definitively colonized thanks to the construction of more than 500 km of canals that made more than 4000 hectares of land suitable for agriculture, mainly for cereal production and vineyards [36]. The village of Maccarese, which developed around the ancient Rospigliosi Castle, San Giorgio Castle, became the nucleus around which production activities as well as social relations, were organized.

Therefore, the landscape of the Arrone basin is characterized, on its lower section, by its agricultural vocation, which brings together different activities of protection and dissemination of an agrarian culture stratified over time and in continuous evolution. Activities of production and conservation, promotion and protection that, linked together in a sustainable approach to the territory, place the rural landscape at the center of new perspectives of integration between ecological, economic and socio-cultural values [37]. The following paragraph concerns the description of the case study process, supported by synthetic tables with the main contract's information related to territorial data, general objectives and specific actions.

### 2.4. The River, Coastal, Landscape Contract Arrone

Because of its naturalistic and historical-cultural value, this rural area is part of a system of protected areas and reserves. In 1996, pursuant to L. 394/1991, the regional reference Law on Protected Areas, the Riserva Statale del Litorale Romano was established, under the double jurisdiction of the municipality of Rome and of that of Fiumicino, each responsible for its territory, for the protection of the undeveloped territory of the coastline and the landscape of the Roman countryside. In 2020, the adoption of the Reserve Management Plan [38] particularly focused attention on the sustainable use and promotion of the rural territory as a central issue for its protection and development.

In the same year, the Office for Small Municipalities and River Contracts of the Lazio region funded the River, Coastal and Landscape Contract Arrone, giving a concrete impulse to the ongoing activities promoted by local associations linked to municipal authorities and environmental organizations (Table 2).

**Table 2.** The River, Coastal, and Landscape Contract Arrone. Source: authors' elaboration (2022) from data from the official web site of the contract available at https://contrattodifiumearrone.it/documentazione-progettuale/ (accessed on 20 July 2022).

| | |
|---|---|
| Territory | Province of Rome<br>Municipalities of Fiumicino, Cerveteri, Anguillara Sabazia, Bracciano, Ladispoli, Manziana and Rome |
| Surface Area | 15.491 hectares |
| Main Water Body | Arrone |
| Natural Protected Areas Included | Riserva Statale Litorale Romano<br>ZPS Tolfetano Cerite Manziate District<br>ZPS Torre Flavia<br>SIC Monte Tosto<br>SIC Sughereta del Sasso<br>SIC Maccgia Grande di Ponte Galeria<br>SIC Bosco di Palo Laziale<br>SIC Caldara di Manziana<br>SIC Macchia di Manziana<br>SIC Monte Paparano<br>SIC Macchia Grande di Focene e Macchia dello Stagneto |
| Promoters | Forum Plinianum Onlus<br>Aries Sistemi S.R.L.,<br>Roman Etruscan Biodistrict<br>Anna Maria Catalano Foundation |

**Table 2.** *Cont.*

| | |
|---|---|
| Partners | Municipality of Cerveteri<br>Consorzio di Bonifica Litorale Nord<br>Leonardo Da Vinci Institute of Higher Education<br>Roman Etruscan Biodistrict Association<br>Naval League–Fiumicino Section<br>Aries Sistemi S.R.L.<br>Nergal S.R.L.<br>Agrivol S.R.L.<br>Assonautica Acqua Interne Lazio e Tevere<br>Anna Maria Catalano Foundation<br>Foedus Foundation<br>Aps Saifo "Sistema archeoambientale Integrato Fiumicino Ostia" |
| Funds | LAZIO REGION<br>Office for Small Municipalities and River Contracts |

The general objectives of the contract are in line with the regional planning policies, foster an integrated approach over a vast area, including: the reduction of water pollution and safeguard of the aquatic environment; the sustainable use of water resources; the redevelopment of environmental and landscape systems pertaining to river corridors; the improvement of the tourist/environmental use of the municipal territories involved; the coordination of the urban and settlement policies of the municipal territories involved; the information sharing and dissemination of the water culture through awareness and education paths on the topic; and the coordination with the hydraulic risk reduction and prevention interventions.

These general objectives have been confirmed with the construction of the Document of Intent (Phase 1) [39] and declined in five assets (landscape and nature, water, quality agriculture, technology and active citizens and enhancement), deepened in the Integrated Knowledge Analysis (Phase 2) [40] and developed in the thematic tables dedicated to the sharing of available documentation, collective confrontation, the formulation of resources, criticalities and proposals and have been integrated with others on coastal erosion, coastal litter pollution, tourism and microplastics to constitute the basis of the Strategic Document (Phase 3) [41] signed in June 2021. Moreover, the explicitation of the Strategic Matrix [42], an attachment to the Strategic Document, evidences the coherence of the general, specific objects and strategic interventions with the multi-level framework of plans and programs already in force in the area (Regional Landscape Territorial Plan-PTPR; Provincial General Territorial Plan-PTPG; General Regulatory Plan-PRG; Management Plan of the Natural Reserve Litorale Romano, Directive 92/43/EEC, Framework Directive 2000/60/EC, Directive 2007/60/EC, Framework Directive 2008/56/EC).

The thematic tables were divided into four main topics (agro-food table; coastal table; water table and tourism table) and led to the definition of the general objectives of the contract (Table 3).

The definition of the Strategic Document shapes and implements the vision that the contract aims at for the entire river basin and represents the main phase of the dynamic and open process initiated with the signing of the Document of Intent. In fact, this first phase of work showed that the community had little knowledge of the landscape and naturalistic values of the territories included in the contract. To this end, a series of initiatives were activated in the territory aimed at involving the citizens in activities related to the knowledge of its territory (Table 4).

**Table 3.** The five assets and general objectives of the River, Coastal, and Landscape Contract Arrone defined in the Document of Intent (Phase 1). Source: authors' elaboration (2022) from data of the official web site of the contract avaiable at https://contrattodifiumearrone.it/documentazione-progettuale/ (accessed on 20 July 2022).

| Assets | General Objectives |
|---|---|
| Landscape and Nature | Protection and enhancement of landscape values<br>Sustainable Tourism<br>Preservation and protection of natural and cultural capital<br>Knowledge of the state and dynamics of the environment |
| Water | Knowledge<br>Coastal area<br>Protection of freshwater quality<br>Protection of water quantity<br>Hydraulic risk<br>Hydraulic risk mitigation<br>Protection and preservation of aquatic environments<br>Emerging issues |
| Quality Agriculture | Sustainable agriculture with low environmental impact<br>Multifunctional agriculture<br>Quality<br>New relationship between water and<br>environmental and food education |
| Technology | Optimization of purification processes<br>Production process waste management<br>Early warnings<br>Emerging issues<br>Production chains |
| Active Citizenship and Valorisation | Participatory planning and citizens involvement<br>Fruition<br>Turistic valorization |

**Table 4.** The initiatives carried out for enhancing the process, according to the five assets of the River, Coastal, and Landscape Contract Arrone. Source: Authors' elaboration (2022) from data of the official web site of the contract available at https://contrattodifiumearrone.it/documentazione-progettuale/ (accessed on 20 July 2022).

| Asset | Projects and Iniziatives |
|---|---|
| Landscape and Nature | "Amarcord" Project<br>Ideas competition "The fresh waters of the Arrone Valley"<br>"The Tree House" Project<br>"Discovering the Villages of Castel di Sasso and Ceri" Project<br>"Project "Children's River Contracts"<br>Proposed projects: "A walk through the centuries' project" and<br>"Wefts of Memory" Project |
| Water | Analysis of the state of the water in areas of high naturalness<br>within the State Natural Reserve Litorale Romano (University of<br>Tuscia)<br>Monitoring of the chemical-physical quantities of aquatic<br>environments included in the WWF Oases (Aries Sisemi srl)<br>Municipality of Cerveteri: two new purification plants<br>Municipality of Fiumicino: one purification plant |
| Quality Agriculture | Rome Agricultural System Project |

**Table 4.** *Cont.*

| Asset | Projects and Iniziatives |
|---|---|
| Technology | X |
| Active Citizenship and Valorisation | SAIL project: "Integrated Archaeoenvironmental System of the Litorale Romano". Participation in the Living Lab Tourism of the Lazio Region Experimental pilot project "Cycling Mirabilia Lazio" |

The synergic and participated work involved in the construction of the Document of Intent (Phase 2) and of the Strategic Document (Phase 3) has therefore led to the Definition of the Action Program (Phase 4) [43] with the articulation of the specific action lines referred to each defined asset for the realization of the River, Coast and Landscape Contract (Table 5).

**Table 5.** The construction of the Action Program (Phase 4) of the River, Coastal, and Landscape Contract Arrone. Source: authors' elaboration (2022) from data of the official web site of the contract available at https://contrattodifiumearrone.it/documentazione-progettuale/ (accessed on 20 July 2022).

| Action Line | Action | Subjects Involved |
|---|---|---|
| 1_Tourism | Integrated and sustainable tourist offer based on the SAIL program "Sistema Archeoambientale Integrato del Litorale laziale Project "Cycling Mirabilia Lazio" | Anna Maria Catalano Foundation SAIFO Committee Visit Ostia Antica APS Local Authorities Coop. LeAli. |
| 2_Active Citizenship and Valorisation | Project "A walk through the centuries" Children's River Contracts" Project | Anna Maria Catalano Foundation Local Authorities UNI Tuscia/DAFNE UNI SAPIENZA/CORIS FOEDUS FOUNDATION IIS Leonardo da Vinci Maccarese IC Marchiafava IC Porto Romano Fiumicino IC Salvo D'Acquisto Cerveteri; Circolo Velico Fiumicino Assonautica Acque Interne |
| 3_Agriculture | Project "Let's transform the information measured in the field into value for the agricultural activity" Vocational training of agricultural operators Environmental and food education in schools | Aries Sistemi srl University of Tuscia—(DAFNE) AREA Science Park Trieste International Centre for Genetic Engineering and Biotechnology Trieste Agenzia Regionale per lo Sviluppo e l'Innovazione dell'Agricoltura del Lazio (ARSIAL SpA) Biodistretto Etrusco Romano PrimoPrincipio Scarl, Alghero (SS) |

## 3. Results

In order to proceed with the assessment of the results, it is necessary to explicitly define the concept of resilience to which the study refers for the process' interpretation and evaluation. The decline of territorial, social and ecological resilience is assumed to interpret and evaluate the methodological process of the contract and its outputs, as a result of actions and initiatives carried out by the implemented phases.

In coherence with the framework described in the introduction (Section 1), this study assumes the declination of Territorial Resilience as the result of a multi-scalar approach and multi-level governance, a negotiated and non-hierarchical cooperation between institutions

that enables an integrated relationship between different actors and facilitates the coherent implementation and coordination of safeguarding and promotion policies concerning the territory. The in-depth knowledge of territorial cultural heritage and the awareness and comprehension of risks and vulnerabilities demonstrate that the effects of disasters due to natural and anthropic hazards can be mitigated by focusing on prevention instead of reaction, improving the relationships between protected sites and resilience for the role of heritage in communities [44].

Social Resilience is intended as the capability of local actors to build networks (multi-level social networks) and define innovative solutions (collaborative planning and participation) to foster flexible and incremental territorial actions. Therefore, the concept of Social Resilience is linked to community and societal governance structures and processes, to the built environment, to human capital and to the natural environment [45,46].

Environmental Resilience (ER) is intended as a multitude of integrated processes aimed at minimizing environmental risks associated with disasters, quickly returning critical environmental and ecological services to functionality, and allowing the learning process to reduce vulnerabilities and risks to future incidents [47]. Therefore, the achievement of environmental resilience is fostered by integrating strategies and actions targeted towards environment and landscape quality, focusing on the relationship between socio-economic fields and planning policies.

According to these premises, the outputs of the process for the construction of the River, Coast and Landscape Contract Arrone can be interpreted following the different implemented phases of the process, from phase 1 to 8 (cfr. Par. 1.3), in order to evaluate the capability of the contract to simultaneously foster territorial, social and environmental resilience.

- Phase 1: Document of Intent. The general objectives of the contract, divided into the five assets (landscape and nature, water, quality agriculture, technology and active citizens and enhancement—Cfr. Par. 2.4 and Table 3) with the consequent definition of activities and initiatives, which combine long-term strategies and short-term actions, supported by top-down institutional policies and bottom-up initiatives (Table 4), result in increasing awareness of territorial values and heritage. The local community was called upon to develop a shared vision by bringing out conflicts, interests and territorial vocations through the definition of the thematic tables, which has contributed to the initiation of a participatory process, which is necessary for the reappropriation of responsibility in the management of water resources. The exhaustive identification of problems and actions expressed in Phase 1: Document of Intent is fundamental for achieving concrete and lasting results in terms of territorial, social and environmental resilience.

- Phase 2: Integrated Knowledge Analysis. This phase, organized around water and landscape and their management as resources, has helped the community recognize the identity of places as a driver for innovation and sustainable change. The thematic tables have seen an increasing level of population involvement on specific issues in their territory, enhancing the rural vocation and the coastal landscape. The four tables on Agro-food, the Coast, Water and Tourism have succeeded in promoting a dialogue between citizens and stakeholders, giving back knowledge of risk and a sense of community. The specific initiatives have demonstrated a wide participation of the communities in the process, in particular the involvement of local educational institutes [48]. The need for empowerment of the youngest through participation in the discovery of the territory has seen the local schools involved in the project "Amarcord" focus, in particular, on the figure of Salvo D'Aquisto, building a network of 28 schools in Italy dedicated to the Italian hero of World War II (Table 4). Knowledge means a profound understanding of local landscapes and cultural values that correspond to new forms of territorial, social and ecological resilience.

- Phase 3: Strategic Document. The multi-level governance approach and the use of "systemic forces", promoters and stakeholders (e.g., research units, schools, associations) represent evidence of improving coherent and quality objectives and interventions

as well as strengthening sustainable activities at the basis of local economies. The Strategic Matrix developed within the Strategic Document ensures the coherence between the contract and the implementation and coordination of safeguarding and promotion plans and programs already in force in the area (Regional Landscape Territorial Plan-PTPR; Provincial General Territorial Plan-PTPG; General Regulatory Plan-PRG; Management Plan of the Natural Reserve Litorale Romano, Directive 92/43/EEC, Framework Directive 2000/60/EC, Directive 2007/60/EC, Framework Directive 2008/56/EC).

- Phase 4: Action Program (AP) The Action Program has set up all the specific interventions aimed at implementing multi-scalar and multi-sectors projects related to integrate water management with a quality approach to river safety, valorization of agricultural productivity, social and economic development, sustainable mobility systems, blue and green infrastructure, nature-based solutions, ecological place-based interventions (Table 5).
- Phase 5: Participatory processes. The sharing of intentions, commitments and responsibilities among authorities, citizens and stakeholders (Table 5) referred to the specific actions of Phase 4 has enabled the definition of the initiatives aimed at improving social involvement and increasing awareness of local identity.

The results of Phase 6: Formal Agreement, Phase 7: Periodic monitoring system and Phase 8: Public information cannot be evaluated yet. The future signing of the agreement will be the institutional prerequisite to consolidate and legitimize the negotiation process, in which the activities carried out by all actors operating in the territory will be inserted.

Although the methodological structure of the procedure, from Phase 1: Document of Intent to Phase 5: Participatory process has been clearly implemented in all its steps and is easily accessible online, allowing a good dissemination of the documents and data and demonstrating a good implementation of Phase 8: Public information, the limits of the procedure are represented by the implementation of Phase 6: Formal Agreement and Phase 7: Periodic monitoring system. This limit is also demonstrated by the overall state of the art of the River Contracts in the Lazio region, which clearly shows that only eight contracts have been formalized to date (Table 1). The monitoring phase, aimed at checking the implementation status of the various phases and actions, the quality of participation and the resulting processes, has not yet been developed in any of the cases cited in Table 1.

Finally, it is possible to state, based on the case study description and evaluation, that the principles of subsidiarity, participation and sustainability of the procedure (Cfr. Par. 1.3) have been respected throughout the whole process as described.

The principle of subsidiarity has been fostered by the horizontal collaboration at local scale between administrations, citizens and associations, by the number of promoters and partners participating in the formalization of Phase 1: Document of Intent (Table 2), by the coherence expressed by the implemented strategic matrix in Phase 3: Strategic Document with the superordinate plans and programs in force in the territory (Cfr. Par. 2.4), and by the different nature of the subjects responsible for the different actions defined in Phase 4: Action Program (Table 5). The principle of participation has been pursued through the thematic tables and initiatives carried out during Phase 1: Document of Intent, Phase 2: Integrated Knowledge Analysis and Phase 5: Participatory Process, as demonstrated by the participation of the citizens in the different projects aimed at increasing awareness and a sense of belonging to their own cultural identity. The principle of sustainability, strongly linked to the concept of resilience, is therefore pursued by the demonstrated procedure's capacity to produce territorial, social and environmental resilience (Section 3).

## 4. Discussion and Conclusions

### 4.1. Discussion

Restoring water landscapes through the implementation of River Contracts, addressing interdependencies between water as a resource and cultural heritage results in a process that communities can take to strengthen their territorial, social and environmental resilience.

River Contracts therefore demonstrate to be a method of governance that promotes the coevolution of territorial systems, projecting them towards the necessary transformations due to environmental and socio-economic changes. The complex ecosystems represented by water landscapes are, in fact, characterized by great opportunities and resources, but also by vulnerability and multiple risks.

The direction taken by this innovative and integrated governance tool, capable of developing synergies and collaborative practices between public and private actors through the construction and regeneration of ecological, infrastructural, cultural and social networks, enables the local community to take care of the existing natural and cultural heritage, reconnecting material and immaterial elements, while redefining the components of a sustainable spatial structure. Therefore, the potentials of River Contracts in safeguarding and improving biodiversity and the multiplicity of benefits they can produce in terms of territorial, social and environmental resilience can contribute to the "good status" of natural, rural and urban ecosystems and, as a consequence, to human well-being.

For this reason, the River Contract's methodology, in accordance with the introductory framework (Section 1), the description given by the case study description (Section 2) and the evaluation (Section 3) returns the relevance of experimenting, at the territorial and local levels, an integrated approach aimed at building territorial, social and ecological awareness and responsibility to face the several effects of climate change.

The effectiveness of the methodology, applied in coherence with the three principles just mentioned, is the reason why, to date, River Contracts are a method practiced by about 3000 municipalities throughout Italy but which nevertheless appears to be full of critical issues, especially in the transition from strategies to projects as well as the operational implementation of the interventions provided by the Action Program.

In the most recent Italian experiences, in the most recent Italian experiences, River Contracts represent the symbol of participatory management of water resources, offering new opportunities to deal with continuous environmental changes, and it is certainly useful, at this still experimental stage of the instrument, to refer to the different approaches, referable to different geographical contexts, which differ in a naturalistic, territorialist-engineering or ecological dominant manner [49].

The naturalistic approach places the water resource at the center of the process as an asset to be protected and for which to envisage a reduction in water quality degradation and the recovery of aquatic habitats. Principles rooted and well recognizable in French experiences, but also in Spain and Switzerland. The territorialist approach, on the other hand, is characteristic of experiences that place the containment of hydrogeological risk and the factors that increase its danger at the center. The experiences conducted in Belgium and later in Germany place the emphasis on the problem of floods and disaster events and aim to achieve their prevention and reduction, including through strong limitations on the buildability of soils in areas affected by flood events.

The ecological approach, characteristic of British experiences, focuses on sustainability and climate change issues, with particular attention to the ecological and landscape improvement of waterways, the enhancement of open spaces and slow mobility, and the protection of natural habitats and ecosystem biodiversity. The issue of sea levels rising is central to the management of the waterscapes to which these contexts belong, a condition to which the Italian situation is very similar because of the strong urbanization that has always affected coastal areas.

*4.2. Conclusions*

The process of metropolization, and its well-known effects on ecosystems, has produced, over time, a loss of natural capital, a loss in the sense of identity and belonging of communities toward their natural and cultural environments, and highlighted the enormous need for resilience-oriented approaches underpinning new restored relationships between landscapes and citizens.

As never before in this historical period marked worldwide by the pandemic, it is fundamental to reorganize people's behavior in feeling, reacting and participating in innovative forms of planning that follow no longer territorial functions but performance. Innovative planning procedures and actions are therefore asked to consistently respond to the objectives of urban and territorial regeneration, promoted at EU and international levels, experimenting the integrated strategies aimed at addressing environmental, social, economic and cultural issues on the basis of new values that refer to the ecological paradigm [50].

Landscape provides us with a range of goods and services, represents an important part of our identity, and has a symbolic meaning because of the equivalence that it has with the context to which it belongs. The multiplicity of demands, perceptions, and uses of landscapes raises challenging questions about how to best plan, design and manage landscapes that are resistant to shocks and adaptive to changes in society and environment.

In this sense, the River Contract method demonstrates its capability to support concerted and participatory management processes with respect to European and national policy frameworks, and to integrate urban and territorial planning with place-based and nature-based interventions for water landscape management.

Therefore, River Contracts represent an innovative tool for the governance of river ecosystems and territories in compliance with *subsidiarity*, *participation* and *sustainability* principles. The shared methodological structure and the top-down and bottom-up approaches have led to a wide evolution and allowed for the identification of river contracts not only as sectoral tools for water resource protection and management but also as catalysts of a new culture of water, recalling the deep interrelationships existing between hydrography, hydrogeology, ecology, sociology, economics, public health and cultural values.

**Funding:** This research activity is funded by Sapienza University of Rome. In particular, the manuscript relates to the activities developed within the framework of the Sapienza University Research (2019) "Rigenerare i territori costieri del Medio Tirreno. Itinerari paesaggistici per comunità resilienti lungo la costa e nelle isole minori del Lazio", principal investigator F. Rossi.

**Data Availability Statement:** Not applicable.

**Conflicts of Interest:** The authors declare no conflict of interest. The funders had no role in the design of the study; in the collection, analyses, or interpretation of data; in the writing of the manuscript; or in the decision to publish the results.

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
