# Peer review of "Method and Practice for Integrated Water Landscapes Management: River Contracts for Resilient Territories and Communities Facing Climate Change"

_urbansci, doi:10.3390/urbansci6040083_

Round 1
Reviewer 1 Report
The article is very interesting and well constructed overall. Nevertheless, we suggest below some areas for improvement.
The introduction is 8 pages long out of the 20 pages of the article, almost half! This is too much! I suggest that the introduction should be summarised, lightened up, to get to the point more quickly. In particular, the developments on the European reference texts (which are a bit tedious as they are presented) and on the notion of ecosystem services (an aspect that is now well known) would benefit from being presented in a more rapid and synthetic manner, so that the reader understands better, and more quickly, what the major issue that the article proposes to study is.
Figure 1: Map to be reworked to improve readability. The shades of blue do not allow the identification of the different Contracts. Adding numbers on the map and in the legend would improve the readability of the map. Also add orientation and some nomenclature elements to facilitate localisations.
page 15: the author mentions "a wide participation of the communities to the process" (l. 572-573). Specify on what elements this assessment is based.
There are still some syntax and spelling errors to be corrected here and there in the article.
Author Response
"Please see the attachment."

Reviewer 2 Report
the paper is very chaotic, it is difficult to understand what the authors mean, it looks like compilations of many legal provisions issued by many organizations dealing with environmental protection.
Author Response
"Please see the attachment."

Reviewer 3 Report
Dear Authors,
The paper entitled "Method and Practice for Integrated Water Landscapes Management: River Contracts for Resilient Territories and Communities facing Climate Change" is a study based on a consistent approach. The theme chosen is original and it is important to carry out research on river contracts, as it is an instrument that can be very useful in favor of sustainability and the participation of users in water management.
However, it is important to make a considerable effort to correct some errors and deficiencies that lower the scientific quality of the paper. Thus, a major revision is proposed. It is necessary to carry out a comprehensive improvement of the paper with the following objectives and criteria:
- River contracts are essentially an instrument of French origin. There are no references that contextualize this aspect, and how Italy has adopted this instrument. For example, in Spain, there are some examples of river contracts but it is not supported by the legal system.
-There is a significant mismatch between the number of pages dedicated to the initial part of the article and the final part, which is the most important. Very little space is devoted to results, discussion and conclusion. A general rethinking of the article is needed.
-The graphic part, in general, needs a major improvement. Figure 1 needs to be rethought so that it is quality thematic cartography.
-A greater effort is necessary in the discussion: comparisons can be made with other case studies. The function of the discussion is to mobilize concepts and own data with those present in the scientific literature. This section is more like previous conclusions than a real discussion.
Authors are encouraged to correct these deficiencies for a high-quality paper.
Best regards,
Reviewer
Author Response
"Please see the attachment."

Round 2
Reviewer 3 Report
Dear Author,
The changes introduced have improved the quality of the paper. The effort made to respond to all of the reviewer's suggestions has been remarkable.
However, two additional changes are proposed:
a) Think about the possibility of including an important bibliographical reference to river contracts:
Brun, A. France's Water Policy: The Interest and Limits of River Contracts. In Globalized Water. A Question of Governance; Schneier-Madanes, G., Ed.; Springer: Dordrecht, The Netherlands, 2014; pp. 139-147.
https://link.springer.com/chapter/10.1007/978-94-007-7323-3_10
b) Maybe removing Figure 1 has not been a good idea. It was interesting to know the geographical context in cartographic suport. Furthermore, your results could be mapped in some way.
Best regards,
Reviewer
Author Response
I have included the Reference suggested.
I have included again Figure 1, better highlighting the geographical context with the regional borders and also highlighting the case study of the Arrone Contract and I have modified the quality, scale and text of Figure 1